# Genome Analysis of a Novel Polysaccharide-Degrading Bacterium *Paenibacillus algicola* and Determination of Alginate Lyases

**DOI:** 10.3390/md20060388

**Published:** 2022-06-09

**Authors:** Huiqin Huang, Zhiguo Zheng, Xiaoxiao Zou, Zixu Wang, Rong Gao, Jun Zhu, Yonghua Hu, Shixiang Bao

**Affiliations:** 1Institute of Tropical Bioscience and Biotechnology, Hainan Institute for Tropical Agricultural Resources, Chinese Academy of Tropical Agricultural Sciences, Haikou 571101, China; huanghuiqin@itbb.org.cn (H.H.); zhengzhiguo52@163.com (Z.Z.); zouxiaoxiao@itbb.org.cn (X.Z.); ever@hainanu.edu.cn (Z.W.); gryt2727@163.com (R.G.); zhujun@itbb.org.cn (J.Z.); 2Zhanjiang Experimental Station, Chinese Academy of Tropical Agricultural Sciences, Zhanjiang 524013, China; 3Hainan Provincial Key Laboratory for Functional Components Research and Utilization of Marine Bioresources, Haikou 571101, China; 4College of Oceanography, Hebei Agricultural University, Qinhuangdao 066000, China; 5Laboratory for Marine Biology and Biotechnology, Pilot National Laboratory for Marine Science and Technology, Qingdao 266071, China

**Keywords:** *Paenibacillus algicola*, genome, polysaccharide lyase, alginate lyase, oligosaccharide

## Abstract

Carbohydrate-active enzymes (CAZymes) are an important characteristic of bacteria in marine systems. We herein describe the CAZymes of *Paenibacillus algicola* HB172198^T^, a novel type species isolated from brown algae in Qishui Bay, Hainan, China. The genome of strain HB172198^T^ is a 4,475,055 bp circular chromosome with an average GC content of 51.2%. Analysis of the nucleotide sequences of the predicted genes shows that strain HB172198^T^ encodes 191 CAZymes. Abundant putative enzymes involved in the degradation of polysaccharides were identified, such as alginate lyase, agarase, carrageenase, xanthanase, xylanase, amylases, cellulase, chitinase, fucosidase and glucanase. Four of the putative polysaccharide lyases from families 7, 15 and 38 were involved in alginate degradation. The alginate lyases of strain HB172198^T^ exhibited the maximum activity 152 U/mL at 50 °C and pH 8.0, and were relatively stable at pH 7.0 and temperatures lower than 40 °C. The average degree of polymerization (DP) of the sodium alginate oligosaccharide (AOS) degraded by the partially purified alginate lyases remained around 14.2, and the thin layer chromatography (TCL) analysis indicated that it contained DP2-DP8 oligosaccharides. The complete genome sequence of *P. algicola* HB172198^T^ will enrich our knowledge of the mechanism of polysaccharide lyase production and provide insights into its potential applications in the degradation of polysaccharides such as alginate.

## 1. Introduction

Complex polysaccharides, including alginate, agar, carrageenan, chitin, cellulose and pectin, etc., are the major components of seaweed cell walls and intercellular spaces, and are generally refractory to degradation [1,2]. Most marine polysaccharides (MPs) are structural components of the cell walls of macroalgae, such as alginate in brown algae (*Phaeophyceae*) and agar and carrageenan in red algae (*Rhodophyceae*). Alginate is widely distributed, mainly in brown seaweeds such as *Laminaria*, *Sargassum* and *Macrocystis* [3]. It is a water-soluble and acidic polysaccharide, consisting of *α*-l-guluronic acid (G) and *β*-d-mannuronic acid (D) in three different arrangements, such as homopolymeric G (PolyG), homopolymeric M (PolyM), alternating GM or random heteropolymeric G/M stretches (polyMG) [4,5]. Alginate, agar and carrageenan are the three typical polysaccharides of marine origin, and are commercially used as thickening, gelling, texturing and stabilizing agents in food, cosmetics and pharmaceuticals [6,7]. On the other hand, algal oligosaccharides, the enzymatic degradation products of MPs, have wide biological activities and used in different fields, such as industry, agriculture and medicine [8,9,10].

The biodegradation of seaweed is crucial to marine ecology and is a key step in material cycles, especially in the carbon cycle [11]. Polysaccharide-degrading bacteria are key players in the global carbon cycle and algal biomass recycling. Many kinds of MP-degrading marine bacteria have been isolated and revealed as the key players in the algal biomass recycling and global carbon cycle, such as *Pseudoalteromonas* [7], *Alteromonas* [12], *Agarivorans* [13], *Paenibacillus* [14], *Vibrio* and *Zobellia* [15,16]. Members of genus *Paenibacillus* produce many kinds of extracellular enzymes, e.g., chitosanase [17], glucanase [18], cellulase/mannanase/xylanase [19], xanthanase [20] and chitinase [21], which can be used in a wide range of industrial fields. However, only a few species in this genus have been reported to possess alginate lyases, namely, *Paenibacillus* sp. LJ-23 from brown algae [22] and *Paenibacillus* sp. strain MY03 [23], *Paenibacillus* sp. S29 [24] and *Paenibacillus* sp. str. FPU-7 [25] from soil.

Alginate lyases can cleave alginate at the hexuronic acid residue sites and release the 4,5-unsaturated hexuronic acid residue at the non-reducing terminus; they attract attention for their broad biotechnological applications, especially in the preparation of biologically active alginate oligosaccharides (AOSs) and the production of biofuels directly from macroalgal biomass [26,27]. According to substrate specificity, alginate lyases are divided into polyM-specific lyases (EC 4.2.2.3), polyG-specific lyases (EC 4.2.2.11) and polyMG-specific lyases (EC 4.2.2), which can degrade polyG, polyM and polyMG blocks of alginate, respectively [28]. Based on the carbohydrate-active enzyme (CAZyme) database (www.cazy.org, accessed on 2 March 2022), alginate lyases are grouped into 14 polysaccharide lyase (PL) families: PL5, PL6, PL7, PL14, PL15, PL17, PL18, and the recently identified families PL31, PL32, PL34, PL36, PL38, PL39 and PL41 [13,21]. CAZyme are the key to promoting carbohydrate catabolism in marine heterotrophic bacteria. Based on genome sequencing and functional annotation, the genomes of several alginate-degrading marine bacteria have been assembled and the CAZyme genes annotated; those such as *Zobellia russellii* and *Z. barbeyronii* [16], *Flammeovirga pacifica* WPAGA1 [1], *Microbulbifer* strain HZ11 [29] and *Paenibacillus* sp. str. FPU-7 [21] have been reported.

Previously, we described *Paenibacillus algicola* HB172198^T^, a novel species isolated from brown algae in Qishui Bay, Hainan, China, with the capability of producing alginate lyase [14]. In the present study, general characteristics of its complete genome sequence are reported, and the genome annotation revealed that diverse CAZymes could degrade various polysaccharides that are constituents of plant and algal cell wall, not just alginate. In addition, we further analyzed the putative polysaccharide lyases (PLs) that are involved in several polysaccharide degradation processes, and investigated the properties of alginate lyase in culture supernatant.

## 2. Results

### 2.1. Screening and Identification of Strain HB172198^T^

Based on the screening results by agar plate method, strain HB172198^T^ from brown seaweed in Qishui Bay, Hainan, China, showed significant alginate lyase activity. Under the action of 1-M CaCl_2_, a gelation reaction white halo and white ring was observed on the plate, which indicated that the strain secreted alginate lyases (Appendix A). The type of calcium ion-dependent reactions on the agar plate containing alginate shows the substrate specificity of alginate lyase [30]. Colonies are circular, light yellow and approximately 1 mm in diameter when grown on a marine agar 2216 (Difco Laboratories, Detroit, MI, USA) plate at 30 °C for 48 h. Cells are Gram-stain-variable, facultatively anaerobic, motile rods (1.8–4.8 × 0.5–0.8 μm) with a polar and a lateral flagella (Appendix A). Phylogenetic analysis of 16S rRNA gene sequences (1474 bp, GenBank No. MG994973) indicated that strain HB172198^T^ belonged to the genus *Paenibacillus*, and the closest phylogenetically related species was *Paenibacillus lemnae* NBRC 109972^T^ (97.6% similarity). Based on the combined phylogenetic relatedness and phenotypic and genotypic features, strain HB172198^T^ was identified as a novel species of the genus *Paenibacillus*, for which the name *Paenibacillus algicola* sp. nov. is proposed. The type strain is HB172198^T^ (=CGMCC 1.13583^T^ = JCM 32683^T^) [14].

### 2.2. Genome Specifics

The complete genome of strain HB172198^T^ was determined and one circular chromosome was obtained, with the GenBank/EMBL/DDBJ accession number CP040396. A total of 87,777 reads were analyzed, with an average read length of 21,386 bp, totaling 1.88 Gb, and 418× coverage depth. Strain HB172198^T^ presents a genome of 4,475,055 bp with chromosomal G + C content of 51.2%. A total of 4182 genes were predicted, including 4001 protein-coding genes and 80 tRNA and 27 rRNA sequences. The general features of the HB172198^T^ genome are shown in Table 1 and Figure 1. The gene functions were classified with COG and KEEG databases, and it was shown that a total of 2950 proteins had clear biological functions, 1842 proteins had KEGG homologous genes and 2946 proteins had COG classification. The most common genes in COG annotation are related to the basic functions of bacterial cells. The highest proportion of genes includes carbohydrate transport and metabolism, amino acid transport and metabolism, transcription, cell motility and secretion.

### 2.3. Genetic Basis of Polysaccharide Degradation

CAZymes are the most important enzymes for polysaccharide degradation. To search for genes related to polysaccharide-degrading enzymes in the genome of *P*. *algicola* HB172198^T^, the carbohydrate-related genes were annotated on the basis of the CAZyme database. Strain HB172198^T^ has 191 CAZymes, including 80 glycoside hydrolases (GHs), 9 polysaccharide lyases (PLs), 53 glycosyl transferases (GTs), 11 carbohydrate esterases (CEs) and 38 carbohydrate-binding modules (CBMs). The PLs are classified into seven families: 7, 8, 12, 15, 31, 38 and 42. The proportion of CAZymes in strain HB172198^T^ is about 4.8%, which is consistent with the statistics of Mann et al. [31]. In general, CAZymes seldom exceed 5% in the genomes of bacteria that specialize in carbohydrate degradation, and typically account for not more than about 2% in most bacterial genomes [32]. The capability of strain HB172198^T^ to degrade various polysaccharides was evident from the annotation of the genome. The genome encodes many kinds of enzymes capable of degrading a diverse range of algal and plant cell wall polysaccharides such as alginate, agar, carrageenan, fucoidan, chitin, xylan, glycosaminoglycan, pullulan, lichenstarch, cellulose, glucan and starch (Table 2). All of these enzymes belong to GHs and PLs.

Further analysis revealed that four of the putative PLs, ORF00773 (accession number: QCT01494.1), ORF02660 (PL15, accession number: QCT03372.1), ORF02668 (PL7, accession number: QCT03380.1) and ORF3334 (PL38, accession number: QCT04045.1), were involved in alginate degradation. The open reading frames (ORFs) of the *orf02660*, *orf02668*, *orf00773* and *orf03334* genes consist of 2319, 936, 3042 and 4833 bp nucleotides; encode 772, 312, 1013 and 1611 amino acids; and contain alginate lyase domains of the PL15, 7, 38 and 38 families, respectively (Appendix A). Aly38A was the first alginate lyase belonging to the PL38 family obtained from *Agarivorans* sp. B2Z047 that has been found to degrade alginate [13]. In the CAZy database, a total of 1383 proteins are classified as PL38 family members. Only two of them, CUL-I and TpPL38A, have been characterized as endo-*β*-1,4-glucuronan lyase, and CUL-I also exhibits alginate lyase activity as well [32,33]. Detailed analysis of the CAZyme information of *Paenibacillus* species was rarely reported. *Paenibacillus* sp. strain MY03 from root soil of cypress had the capability of metabolizing polysaccharides of marine algae and animals. Various polysaccharidase genes related to seaweed degradation were found in its genome, including a glucoamylase, a mannanase, an alginate lyase, 3 putative agarases, 4 glucanases and 10 xylanases [23]. However, further analysis based on the dbCAN server (http://cys.bios.niu.edu/dbCAN2, accessed on 8 May 2022) revealed that five of the putative PLs were involved in alginate degradation, namely two PL6, one PL14 and two PL15, which was inconsistent with Liu et al. [23]. No genomic data for other *Paenibacillus* species with the ability to degrade alginate were found on the NCBI webpage. To further confirm the attribution of the alginate lyases, a phylogenetic tree was constructed according to the amino acid sequences of the four enzymes and other reported alginate lyases. As shown in Figure 2, ORF02660 and ORF02668 were clearly located in the clade with the PL15 and PL7 families, respectively; moreover, ORF00773 and ORF03334 were located in the clade with the PL38 family and formed a distinct branch, which was consistent with the results predicted by the CAZy database.

Additionally, two agar-degrading genes belonging to the GH50 and GH86 families were predicted, whose members are known in the CAZy database for *β*-agarase and porphyrinase activities [34]. Agarose is generally hydrolyzed by GH86 family *β*-agarase to generate neoagarotetraose and neoagarohexaose, while neoagarooligosaccharide is generally hydrolyzed by GH50 family *β*-agarase to generate neoagarobiaose [35]. Both of the *β*-agarases in strain HB172198^T^ would degrade agar to neoagarobiaose. The *ι*-carrageenase identified in strain HB172198^T^ belongs to the GH82 family. It is reported that all members of the GH82 family demonstrate carrageenase activity against *ι*-carrageenan.

Furthermore, two cellulase genes were identified in strain HB172198^T^. One *β*-1,4-endo-glucanase (GH9 family) and four *β*-glucosidases (GH3 family) were found in the genome of strain HB172198^T^. Generally, cellulose hydrolysis is achieved by the synergistic action of endo-glucanase, exo-glucanase and *β*-glucosidase.

Some bacteria were reported as degraders of xylan and called xylanases. *P*. *algicola* HB172198^T^ can also degrade xylan, which is a group of hemicelluloses found in plant cell walls and some algae [36]. Seven putative xylanases exist in strain HB172198^T^, including three putative endo-1,4-*β*-xylanases (GH10 and GH11 families), and four putative xylan 1,4-*β*-xylosidases (GH43 and GH52 families). Two endo-1,4-*β*-xylanases are grouped with the GH10 family, which are modular enzymes. CBM9 and CBM22 modules were observed and the modular structure of xylanases facilitates the binding of enzymes to substrates.

A total of five *α*-amylase and two pullulanase genes were identified. One *α*-amylase is classified in the GH2 family and three in GH13 families, whereas the fifth one is a modular enzyme belonging to the GH13 family that is appended with carbohydrate-binding modules CBM34. The two pullulanases are both modular enzymes belonging to the GH13 family with additional carbohydrate-binding modules CBM48 and CBM41.

Besides the enzymes responsible for degrading these above-mentioned polysaccharides, numerous other saccharide-degrading enzymes were predicted from the *P. algicola* HB172198^T^ genome, including *α*-l-fucosidase, lichenase, glucan endo-1,3-*β*-d-glucosidase, *α*-glucosidase, *α*-galactosidase, *β*-galactosidase, *α*-l-rhamnosidase, *α*-mannosidase, arabinanase and *α*-phosphotrehalase. These masses of CAZymes comprise a complex system for carbohydrate catabolism in strain HB172198^T^.

### 2.4. Test of Carbohydrate Utilization

The OD_600_ of the fermentation broth was measured after 48 h incubation at 180 rpm and 30 °C on an orbital shaker. Cell growth represents that the tested carbohydrates can be utilized by the strain as a carbon source for its growth. Excitingly, all the tested carbon sources could be utilized for the growth of strain HB172198^T^ as the sole carbon source, such as alginate, carrageenan, agar, chitin, starch, cellulose, hemicellulose, xylan, xanthan, sucrose, lactose, maltose, glucose, rhamnose, fructose, xylose and arabinose (Appendix A). For the complex polysaccharides, strain HB172198^T^ grew best on alginate (OD_600_ = 1.152), followed by chitin, carrageenan, xanthan, xylan, starch, agar, hemicellulose and cellulose. Therefore, the results indicate that strain HB172198^T^ harbors powerful enzymes for utilizing various simple and complex carbohydrates, which would enable it to adapt to various environments.

### 2.5. Enzymatic Properties of Alginate Lyase

Fermentation was carried out using the optimal fermentation medium and condition; the supernatant was obtained by centrifugation, with the alginate lyase activity of 152 U/mL determined by the ultraviolet absorption method. The culture supernatant was precipitated by saturation of ammonium sulfate (80%) to prepare extracellular alginate lyase. The specific activity was increased to 1839 U mg/L from the initial 152 U mg/L, with a yield of 12.1.

The effects of pH and temperature on the activity of the partially purified alginate lyase were examined. As shown in Figure 3A, the alginate lyase from strain HB172198^T^ exhibited maximum enzymatic activity at 50 °C. Nearly 80% of the highest activity was manifested at the temperature range of 40–60 °C, while almost no detectable activity was observed at 4 °C and 80 °C. The thermostability of alginate lyases was determined at a temperature ranging from 4 to 90 °C (Figure 3B). The alginate lyases were relatively stable at temperatures lower than 40 °C; approximately 100% of the activity was maintained after incubation at less than 40 °C for 1 h. As the temperature rose above 40 °C, the activity declined dramatically; and the vast majority of the activity was lost above 70 °C. As shown in Figure 3C, the activity was the highest at pH 8.0, and above 70% of the maximum activity when the pH value was between 7.0 and 9.0. The activity was the most stable at pH 7.0, above 90% of the activity was retained at pH 6–8 and about 60% of the activity at pH 3.0 and pH 9.0 (Figure 3D). These results indicate that the alginate lyases have good pH stability.

Results of various metal ions and compounds on alginate lyase showed that the ions (1-mM) of Ca^2+^, Mg^2+^, NH_4_^+^ and Fe^2+^ displayed a slightly promoted effect on the enzyme activity, but Zn^2+^, Mn^2+^ and urea had weak inhibitory effect. Ba^2+^ showed some inhibitory effect with 80.3% of the relative activity, and EDTA directly reduced the enzymatic activity to 27.1% of the control group.

The substrate specificity was detected by measuring the increased absorbance at 235 nm of the unsaturated uronic acids that were generated from the oligomers via a *β*-elimination reaction. According to the results of the substrate specificity assay, the alginate lyases from strain HB172198^T^ exhibited higher activity with polyM than with alginate and polyG (Appendix A). The ability to degrade polyG was only 56.0% of that of alginate, while the ability to degrade polyM was 2.52 times that of alginate. Obviously, the enzymes could act more significantly on polyM than polyG and sodium alginate. This suggested that the enzymes were suitable for the production of mannuronate oligosaccharides from polyM blocks, and the production of oligosaccharides from sodium alginate.

### 2.6. Enzymatic Degradation of Sodium Alginate

The extent of polysaccharide degradation was determined using the partially purified alginate lyase, the contents of reducing and total sugars were monitored and the average DP of the alginate fragments was calculated. Under the optimized conditions of 1.2% sodium alginate, 18.60 U/mL enzyme, pH 7.0 and 45 °C for 36 h, the yield of reducing sugar and the average DP are shown in Figure 4. The action of the enzyme solution on sodium alginate resulted in the release of reducing sugars, with a 50% increase in reducing sugars during 12 h incubation. With the increase of enzymatic hydrolysis time, the content of reducing sugar increased rapidly, and the average DP of AOS decreased rapidly. After 36 h, both of them were gradually stabilized, the reducing sugar content reached a maximum value of 34.0 mg/L, and the average DP of oligosaccharides reached a minimum value of 14.2. After hydrolysis for 36 h, there was no obvious increase in the yield of reducing sugars, partially because the enzyme lost its activity after being incubated at 45 °C for a long time.

TLC was used to detect AOS with low DPs. As the hydrolysis process proceeded, AOS with different DPs appeared (Appendix A). Lanes 1-8 mean the digested samples with enzymolysis times of 0, 2, 6, 12, 24, 36, 48 and 60 h, respectively. When incubated for 2 and 6 h, small amounts of low DP oligosaccharides began to appear. When incubated for 24–48 h, AOS with small DP showed higher content, which was consistent with the detection results of reducing sugars. In addition, the oligosaccharide content decreased to some extent after 60 h of incubation. Results showed that the AOS with various low degrees of polymerization (DPs 2–8) were continuous. In general, only oligosaccharides below DP8 can be developed under the TLC conditions employed. There was no monosaccharide in the TLC results, indicating that the four alginate lyases produced by strain HB172198^T^ were all endolytic lyases.

## 3. Materials and Methods

### 3.1. Materials and Strains

The polysaccharides of sodium alginate, carrageenan, agar, cellulose, chitin, starch, glucan, hemicellulose, xylan and xanthan were purchased from Sangon (Shanghai, China). PolyM and polyG (purity > 97%) were purchased from Qingdao Haizhida Biotech Co., Ltd. (Qingdao, China). Other chemicals and reagents used in this study were of analytical grade.

### 3.2. Screening and Identification of Strain HB172198^T^

Brown seaweed samples were collected from Qishui Bay, Hainan, China (19°38′6″ N, 111°0′21″ E). For the isolation of spore-forming bacteria, the mashed sample was incubated at 80 °C for 15 min to kill any vegetative cells. Then, suspension liquid was serially diluted with sterile saline water and spread on modified 2216E agar (MA; Difco) supplemented with 0.5% sodium alginate. The alginate lyase activity was preliminarily screened with the agar plate method using 1-M calcium chloride as the enzyme-producing indicator. The substrate specificity of alginate lyase can be determined by discriminating between the types of gelation (i.e., halo or ring formation) caused by the interaction between calcium ions and depolymerized alginates [30]. The bacterial isolate HB172198^T^ was picked and identified using a polyphasic approach [14].

### 3.3. Genome Sequencing and Annotation

Cells of strain HB172198^T^ were cultured overnight in 2216E medium (MB; Difco). DNA was extracted using TIANamp Bacteria DNA Kit (Qiagen, DP302) following the manufacturer’s protocol. The quality and size of genomic DNA were determined by 0.8% agarose gel electrophoresis, NanoDrop 2000 (Thermo Scientific, Waltham, MA, USA) and Qubit version 2.0 fluorometer (Invitrogen, Carlsbad, CA, USA). A high-quality genome sequence of strain HB172198^T^ was obtained using the PacBio RSII system (Pacific Biosciences, Menlo Park, CA, USA) and Illumina X10 (San Diego, CA, USA) at the Chinese National Human Genome Center (Shanghai, China). The 10 kb library of inserts was constructed by using DNA Template Prep Kit 4.0 and sequenced on the Pacbio RSII system (Pacific Biosciences, Menlo Park, CA, USA). The pair-end library of the 300 bp insert was constructed by using the TruSeq^TM^ DNA Sample Prep Kit-Set A and sequenced on Illumina X10 (Illumina, San Diego, CA, USA). The clean data from Illumina sequencing were corrected for the assembly of PacBio by HGAP v. 23 to generate one contig without gaps. Protein-coding sequences were predicted with Glimmer version 3.02 software [37]; transfer RNA (tRNA) and ribosomal RNA (rRNA) were predicted with tRNAScan [38] and RNAmmer [39]. Functional annotation of the predicted protein-coding genes was performed against the non-redundant protein (NR) database and the GO [40], COG [41], KEGG [42] databases, respectively. The CAZymes and carbohydrate-binding modules were predicted using the BLASTP and CAZy database (http://www.cazy.org/, accessed on 25 February 2022) [34]. The signal peptide was predicted using the SingalP server (https://services.healthtech.dtu.dk/service.php?SignalP-5.0, accessed on 26 February 2022) [43]. The theoretical isoelectronic point (pI) and molecular weight (Mw) were predicted online (http://web.expasy.org/protparam/, accessed on 26 February 2022). The protein domain prediction was performed with the Simple Modular Architecture Research Tool (SMART) (http://web.expasy.org/protparam/, accessed on 26 February 2022). The neighbor-joining phylogenetic tree was generated based on the reported alginate lyases using MEGA version 7.0 [44].

### 3.4. Utilizing Abilities of Carbohydrate

The alginate-degrading strain HB172198^T^, which was classified into the genus *Paenibacillus*, was further investigated regarding its ability to utilize a range of different carbohydrates as sole carbon source in a marine minimal medium (MMM; *w*/*v*: 0.5% (NH_4_)_2_SO_4_, 0.2% K_2_HPO_4_, 2% NaCl, 0.1% MgSO_4_·7H_2_O, 0.001% FeSO_4_·7H_2_O, pH 7.5), containing 0.5% of each mono-, oligo- or polysaccharide substrate. The following carbohydrates were used as sole carbon source: polysaccharides such as sodium alginate, carrageenan, agar, colloidal chitin, starch, cellulose, hemicellulose, xylan and xanthan; disaccharides such as sucrose, lactose and maltose; monosaccharides such as glucose, rhamnose, fructose, xylose and arabinose. After incubating for 48 h on an orbital shaker at 180 rpm and 30 °C, the OD_600_ was detected to determine whether the strain could utilize the tested carbohydrate as the carbon source for its growth.

### 3.5. Detection of Alginate Lyase Activity and Enzymatic Properties

Strain HB172198^T^ was propagated at 30 °C and 180 rpm using the optimized liquid medium, which contained: sodium alginate 7.50 g/L, tryptone 13.57 g/L, NaCl 29.75 g/L, MgSO_4_·7H_2_O 0.08 g/L, pH 7.0. After 36 h of incubation, cells were removed by centrifugation at 10,000 rpm, 4 °C for 15 min. The supernatant was taken as the crude enzyme solution to detect alginate lyase activity with the ultraviolet absorption method [45]. One unit of enzyme activity was defined as an increase of 0.01 in absorbance per min at 235 nm.

In the following, the operating temperature of alginate lyase was maintained at 4 °C unless otherwise stated. The cell-free supernatant was precipitated by 80% saturation of ammonium sulfate and kept overnight. The precipitated protein was collected by centrifugation (10,000 rpm, 30 min) and dissolved in 0.05 M phosphate-citrate buffer at pH 7.0. This enzyme solution was dialyzed in a dialysis bag (MWCO: 12,000–14,000 Da) against the same buffer four times, changing the dialysis buffer every six hours. The dialyzed supernatant was used as a partially purified alginate lyase for the enzymatic property.

To determine the optimal temperature, the enzyme activity was measured at various temperatures (4 °C, 30–80 °C at 10 °C increments) and pH 7.0. To test thermal stability, the enzyme was preincubated at various temperatures and pH 7.0 for 1 h. To determine the optimal pH, the enzyme activity was measured at various pHs using citrate-phosphate (pH 3.0–7.0) and Tris-HCl (pH 8.0–9.0) buffers at 40 °C. To test pH stability, the enzyme was preincubated at various pHs (pH 3.0–9.0, with increments of 1) at 4 °C for 24 h. The highest activity was taken as 100%. To determine the metal ions and compounds, the enzyme activity was measured at 1-mM KCl, CaCl_2_, NH_4_Cl, FeSO_4_, MgCl_2_, ZnSO_4_, BaCl_2_, MnSO_4_, urea and ethylenediamine tetraacetic acid (EDTA), respectively. The enzyme activity without the treatment or addition of extra substances was defined as 100%. All reactions were performed in triplicate. After each treatment, the enzyme activity was estimated by measuring the absorbance at 235 nm.

The enzyme activity assays of sodium alginate, polyM and polyG were defined for investigating the substrate specificity. The amount of yielded unsaturated uronic acid was monitored by recording the absorbance of the reaction mixture at 235 nm, using sodium alginate as the reference (100%) [46].

### 3.6. Preparation and Detection of the Enzymatic Degradation of Alginate

To elucidate the effect of the enzymes concerning polysaccharide, alginate degradation was performed with 12 g/L sodium alginate in 50 mM phosphate buffer (pH 7.0) as a substrate. The experiments were carried out under the optimized conditions of 1.2% sodium alginate, 18.60 U/mL enzyme, pH 7.0 and 45 °C. At intervals (0, 2, 6, 12, 24, 36, 48 and 60 h), aliquot samples (10 mL) were taken, boiled for 10 min to denature the enzymes and the polysaccharides were precipitated overnight with three times the volume of ethanol. After centrifugation at 10,000 rpm for 15 min, the supernatant was taken, lyophilized and then redissolved in distilled water to reduce sugar, total sugar and TLC testing. The content of reducing sugar was determined by using 3,5-dinitrosalicylic acid (DNS) colorimetry [47]. The content of total carbohydrate was determined with the phenol-sulfuric acid method described by Doubois et al. [48]. The average DP of the alginate fragments was calculated by dividing the total sugar content by the reducing sugar content. To determine the enzymatic depolymerization pattern of sodium alginate, AOSs were also measured with the TLC method on a silica gel high-performance TLC plate (Merck, Germany) with the solvent system (1-butanol/formic acid/water 4:5:1) and visualized by heating at 110 °C for 5 min after spraying with 10% (*v*/*v*) sulfuric acid in ethanol [49]. The guluronic acid sodium salt monomers, trimers and pentamers (1 mg/mL) (Qingdao Bozhi Huili Biotech, Qingdao, China) were used as standards.

## 4. Conclusions

In this work, the genome of a novel alginate lyase-producing marine bacterium, designated *Paenibacillus algicola* HB172198^T^, was sequenced. The assembled fine genome contains 4,475,055 bp with G + C content of 51.2%. Among 4182 genes, 4001 protein-coding genes and 80 tRNA and 27 rRNA sequences were predicted. Analysis of nucleotide sequence of predicted gene using the CAZymes Analysis Toolkit indicated that strain HB172198^T^ encodes 191 CAZymes, including 80 glycoside hydrolases, 11 carbohydrate esterases, 9 polysaccharide lyases and 38 carbohydrate-binding modules. In addition, abundant putative enzymes involved in degrading polysaccharide were found, including alginate lyases, agarase, carrageenase, cellulase, xylanases, amylase, pullulanase, chitinase, xanthanase, fucosidase, lichenase, glucanase, etc. The crude extracellular alginate lyase activity of strain HB172198^T^ reached 152 U/mL using the optimized liquid medium at 30 °C and 180 rpm for 36 h. The average DP of oligosaccharide-degrading sodium alginate was maintained at about 14.2, and the oligosaccharide components of DP2-DP8 were present as well. Our results show that *Paenibacillus algicola* HB172198^T^ is therefore a source of potential MP-degrading biocatalysts for biorefinery applications and oligosaccharide preparation.

## Figures and Tables

**Figure 1 marinedrugs-20-00388-f001:**
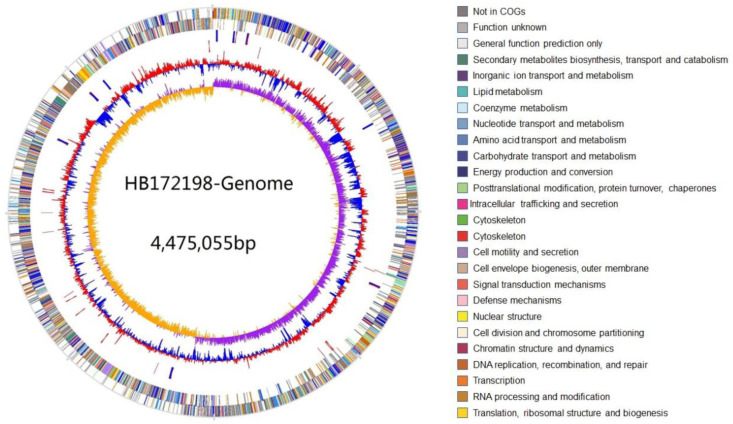
Graphical map of strain HB172198^T^ genome. From the outside to the center: The outer two circles illustrate predicted coding sequences on the plus and minus strands, respectively, colored by functional categories according to COG classification. The 3rd circle displays tRNA (red) and rRNA (blue). The 4th circle represents mean centered G + C content of the genome (red—above mean; blue—below mean). The 5th circle (innermost) represents GC skew (G − C)/(G + C) calculated using a 2 kb window in steps of 1 kb.

**Figure 2 marinedrugs-20-00388-f002:**
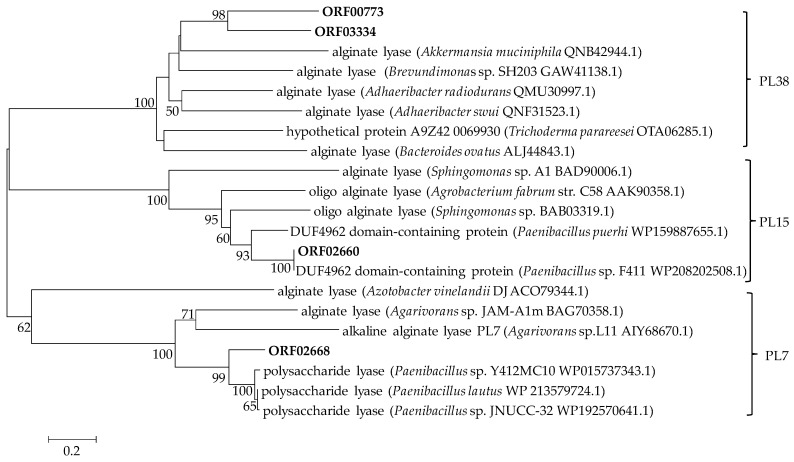
Neighbor-joining molecular phylogenetic tree of alginate lyases belonging to PL7, PL15 and PL38 families based on predicted amino acid sequences. Bootstrap values (1000 replicates) are shown as percentages at each node for values. The scale bar represents 0.2 nucleotide substitutions per position. Putative alginate lyases of strain HB172198^T^ are highlighted in bold.

**Figure 3 marinedrugs-20-00388-f003:**
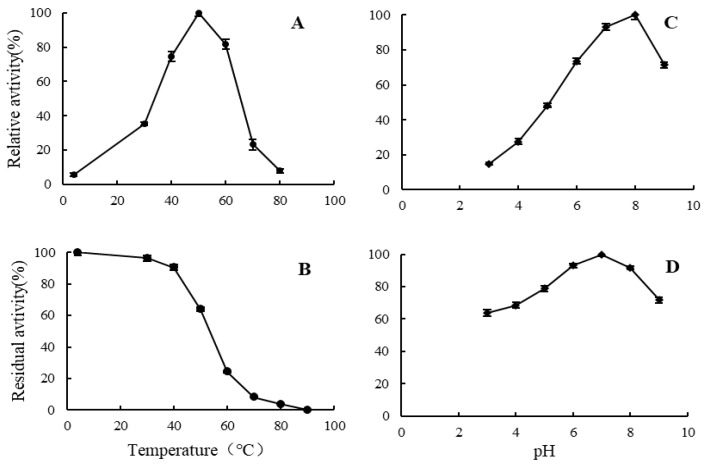
The biochemical characteristics of alginate lyases. (**A**) Effect of different temperatures on the activity (4–80 °C). (**B**) Effect of different temperatures on the stability (4–90 °C). (**C**) Effect of different pH levels on the activity (pH 3–9). (**D**) Effect of different pH levels on the stability (pH 3–9). The highest activity was taken as 100%. Data are given as the means ± standard deviation, *n* = 3.

**Figure 4 marinedrugs-20-00388-f004:**
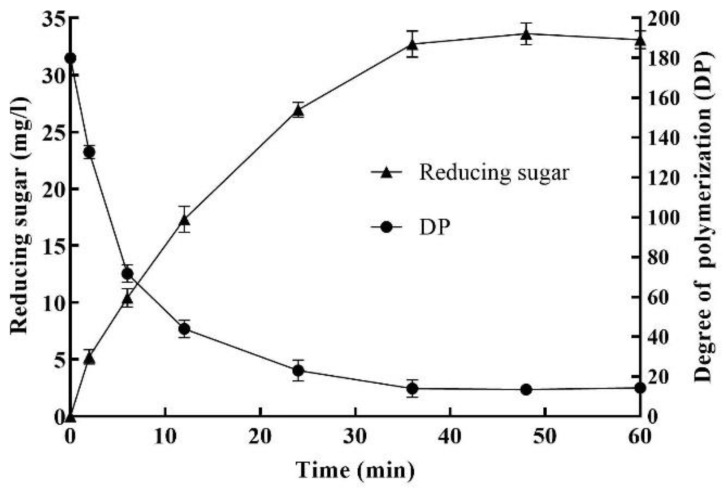
Effect of enzymolysis time on the enzymatic hydrolysis of sodium alginate under the optimized conditions of 1.2% sodium alginate, 18.60 U/mL enzyme, pH 7.0 and 45 °C.

**Table 1 marinedrugs-20-00388-t001:** General features of the *P*. *algicola* HB172198^T^ genome.

Category	Number
Genome size (bp)	4,475,055
G+C content (%)	51.2%
Total genes predicted	4182
Protein-coding genes	4001
tRNA genes	80
rRNA genes	27
5S rRNA	9
16S rRNA	9
23S rRNA	9
ncRNAs	4
Pseudo genes (total)	70

**Table 2 marinedrugs-20-00388-t002:** Diverse genes related to polysaccharide degradation identified in the genome of *P*. *algicola* HB172198^T^.

Catabolic Enzymes	Enzyme Family	No. of Enymes
Alginate lyase	PL7	1
	PL15	1
	PL38	2
*β*-Agarase	GH50	1
	GH86	1
*ι*-Carrageenase	GH82	1
*β*-1,4-Endo-glucanase	GH9	1
*β*-Glucosidase	GH3	4
*α*-Amylase	GH2	1
	GH13	3
	GH13|CBM34	1
Pullulanase	CBM48|GH13|CBM41	1
	CBM41|CBM41|CBM48|GH13|CBM41|GH13	1
Lichenase	GH16	1
Endo-1,4-*β*-xylanase	CBM22|GH10|CBM9	1
	CBM22|GH10|CBM9|CBM9	1
	GH11	1
Xylan 1,4-*β*-xylosidase	GH43	3
	GH52	1
*α*-Glucosidase	GH4	1
	GH13	1
	CBM34|GH13	1
Heparinase	PL12	1
Chitinase	GH18	1
*α*-L-fucosidase	GH29	1
Glucan endo-1,3-*β*-D-glucosidase	CBM54|GH16|CBM4|CBM4|CBM4|CBM4	1
Glycosaminoglycan polysaccharide lyase	PL8	1
*α*-Galactosidase	GH4	1
*β*-Galactosidase	GH36	1
	GH2	4
	GHnc|CBM66	1
*α*-L-rhamnosidase	GH78	3
*α*-Mannosidase	GH38	2
	GH125	1
*α*-Phosphotrehalase	GH13	1
Arabinanase	GH117	1

## Data Availability

Not applicable.

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
