# Peer review of "Genome Analysis of a Novel Polysaccharide-Degrading Bacterium Paenibacillus algicola and Determination of Alginate Lyases"

_marinedrugs, 2022, doi:10.3390/md20060388_

Round 1
Reviewer 1 Report
This paper is a ‘continuation’ of the discovery of the particular Paenibacillus algicola strain HB172198T (discovered/reported previously by the authors). The genome has been deposited and the work has merit, but also some shortcomings that must be addressed/corrected before the paper can be endorsed for publication.
- Introduction: There are some slightly wrong reference citations and omissions, and too many self-citations to provide a credible, objective introduction. The following items are critical:
1a) Eg ref 1 refers to the discovery of secondary walls and lignin within cells of the intertidal red alga Calliarthron cheilosporioides; you need to find another reference for the very first statement. 1b) Refs. 5-6 cannot be used to support the statement about commercial uses (ref. 5 concerns agarase, and ref 6 is another paper about alginate lyase discovery). For the industrial uses it would be better to refer to a paper that considers the application potential and e.g. the proper gelation details of alginate (eg 10.1016/j.tifs.2021.01.052).
1c) To have proper focus, it would also be recommended to weed out citations of agarase, eventhough you refer to “MP-degrading”, but agarose is very different from alginate, and the enzymatic degradation of agar and agarose is different – agarases, alpha and beta-agarases are hydrolases, not lyases.
1d) For the “carbon-cycle” statement, line 55-57, there is no reference. I suggest you consult www.biorxiv.org/content/10.1101/2022.03.04.483023 to underpin and specify the statement.
1e) More specific details about alginate lyases would be welcome. It could for example be mentioned that alginate lyases (EC 4.2.2.x) are categorized as Polysaccharide Lyases, and they belong to 12 different families in the CAZy database.
- Results and methods.
2a) Line 88-89: This statement is only valid if you use polyM and polyG in the plates. That needs to be specified.
2b) The methods to select for alginate lyase activity are not well chosen. Lyases work by a beta-elimination mechanism on the uronic acid backbone and leave behind a “signature” Δ4,5 unsaturated bonds at the nonreducing end of one of the cleavage products. Hydrolases also produce reducing ends as you have measured (DNS method, line 342), and you also used TLC. It would be more relevant to measure the distinct formation of Δ4,5 unsaturated bonds they absorb in the low UV.
2c) General results: esp. Fig 3 and Fig 4: Assays in general: Alginate is a mix of MG structural moieties. For proper analysis you need to include assays on polyG and polyM –
2d) Line 337 and elsewhere How is an alginate lyase unit defined?
2e) You need to expand the discussion in two ways
2e, 1): How do these data compare to the action of more wellknown alginate lyases – it is required to include a discussion of the performance of these alginate lyases vs the available data on PL7 alginate lyases from Flavobacterium and/or Sphingomonas; the alginate lyase from Sigma-Aldrich could be considered for comparison. That enzyme is most likely from Flavobacterium multivorum and it has been characterized on true alginate and on polyG and polyM and its thermal stability has been published (DOI: 10.1039/c6ra06669k); at least compare the characterization Fig 3 to available data on that enzyme as a benchmark.
2f) line 349: “gurouronic acid”? Should this be guluronic acid?
- Conclusions
3a) Line 362-364: 152 U/ml should be clarified (conditions, protein concentration?).
Author Response
Dear editor,
Thank you and the reviewers for the valuable suggestions provided and we have carefully addressed all the comments.

Reviewer 2 Report
This paper reported genome of polysaccharide-degrading bacteria and characterization of its alginate lyases. CAZymes are very interesting enzyme in field of biotechnology, so CAZyme secretion bacteria is useful sources in this field. In this point, this report are provide useful information.
Comments:
How many alginate lyases were found in the genome, and what enzymes activity tested defined alginate lyases gene? How is it connected to alginate lyases, which has been tested for enzyme activity, and various CAZyme in the genome? This paper reported whole genome information of new species and characterization of secreted alginate lyases, discussion of the validity of the two information is needed.
The method need more detail, especially enzymatic properties.
Figure legends need detail information, ex. Figure 4, I can't understand this figure. Is this the effect of substrate concentration?
Author Response

(The authors gave the same response as above.)

Reviewer 3 Report
Dear Editor,
Marine drugs, MDPI,
Huang H, and co-authors report on complete genomic information and CAZymes annotation of Paenibacillus algicola from the brown seaweed. They find out, the strain was able to utilize the different polysaccharides and highlighted on the alginate lyases. Based on their previous finding (report) they focused on investigating the properties of alginate lyases in supernatant. I have some suggestions to enrich the manuscript before publication.
- The phylogenetic analysis of 4 OFR that predicts them to be PLs and to be alginate lyase. Only based on this prediction it is difficult to suppose all ORF to be exact alginate lyase. On the other hand, Table S1 shows the different size and different isoelectric pH for different gene. How they can be analyze together to show the enzymatic properties of alginate lyase (2.5 - result and discussion section), line 202 ? It would be better to characterized the individual gene by heterologous expression and compare their biochemical properties.
- Las line of introduction, ‘Our work will enrich the understanding of PL-producing mechanism of Paenibacillus ’ I think it may be difficult to conclude the above sentence by screening of only one species of Paenibacillus. Some comparative analysis on PLs production or CAZymes distribution between different Paenibacillus species available based on specific habitat or as general, may add the significance.
- Line 128, authors describe the CAZyme information in algicola. Please add more comparison and discussion with other Panibacillus sp. CAZymes statistics.
- In the results and discussion section 2.4 It will be better to perform growth pattern in the presence of different polysaccharides and measure the activity of alginate lyases at their log phase. Please supplement the growth pattern of strain in presence of different polysaccharides and only in minimal media.
- All the polysaccharides were tested such as sodium alginate, carrageenan, agar, colloidal chitin, starch, cellulose, hemicellulose, xylan and xanthan; disaccharides such as sucrose, lactose and maltose; monosaccharides such as glucose, rhamnose, fructose, xylose and arabinose. However, there is discussion about sodium alginate, it did not cover other polysaccharides information related. Therefore, further discussion of polysaccharides should be supplemented.
- From the result of TLC analysis (Figure S3), there is no information related to numbers from 1 to 8, and it should be supplemented.
- There is no discussion about alginate lyase highlighting that it is important for biotechnological application. Besides that, comparison can be done with other already characterized alginate lyase. You can refer to this paper (Lang, Yinzhi, et al. "Applications of mass spectrometry to structural analysis of marine oligosaccharides." Marine Drugs 12.7 (2014): 4005-4030)
- If possible, confirmational analysis of oligosaccharides through ESI-CID MS/MS can be done. You can refer to this paper (Lang, Yinzhi, et al. "Applications of mass spectrometry to structural analysis of marine oligosaccharides." Marine Drugs 12.7 (2014): 4005-4030)
- Table 1 General features of P. algicola HB172198T genome (The number of rRNA genes is missing in the table).
- The spelling of neagarobiaose, neagarotetraose, neagarohexaose should be spell correctly.
- Please mention the Preliminary Data tested on the carbon sources which could be utilized for the growth of strain HB172198T as sole carbon source such as alginate, carrageenan, agar, chitin, starch, cellulose, hemicellulose, xylan, xanthan, sucrose, lactose, maltose, glucose, rhamnose, fructose, xylose and arabinose.
- If possible, according to the mode of action and the type of alginate oligosaccharides (DP of the alginate oligosaccharides) produced by enzymatic hydrolysis of sodium alginate mention the type of enzyme is that (endolytic or exolytic).
Author Response

(The authors gave the same response as above.)

Round 2
Reviewer 2 Report
The authors have revised all my comments.